# Complex Eigenvalue Analysis of Multibody Problems via Sparsity-Preserving Krylov–Schur Iterations

Dario Mangoni *,†, Alessandro Tasora †  and Chao Peng †

Department of Engineering and Architecture, University of Parma, Parco Area delle Scienze 181/A, 43124 Parma, Italy
* Correspondence: dario.mangoni@unipr.it
† These authors contributed equally to this work.

**Abstract:** In this work, we discuss the numerical challenges involved in the computation of the complex eigenvalues of damped multi-flexible-body problems. Aiming at the highest generality, the candidate method must be able to deal with arbitrary rigid body modes (free–free mechanisms), arbitrary algebraic constraints, and must be able to exploit the sparsity pattern of Jacobians of large systems. We propose a custom implementation of the Krylov–Schur method, proving its robustness and its accuracy in a variety of different complex test cases.

**Keywords:** modal analysis; eigenvalues; multibody; damped modes; sparse eigenproblem





## 1. Introduction

Equations of the motion of multibody systems are highly nonlinear in general, but there are cases where one is interested in a linearization of such equations as a way to study the effects of perturbations around a given configuration. To this end, being able to compute the eigenvalues and eigenvectors of the linearized model is of fundamental importance, and it is not limited to the conventional methods of a modal analysis.

For instance, among other applications, eigenvectors can be used to perform a component mode synthesis, also known as *modal reduction*, that is an effective approach which turns a complex system into a surrogate model with a smaller set of coordinates, hence obtaining faster simulations [1–3]. Another application that requires the computation of eigenpairs is the stability analysis of dynamic systems, for instance, the aeroelastic stability of helicopter blades, wind turbines and other slender structures. In this case, one needs to implement a complex-valued eigenvalue problem, where the imaginary and real parts of the eigenvalues give an indication of the damping factor and, consequently, an indication about the impending instability [4,5]. Finally, we can mention that, in the field of control theory, often a state-space representation of the linearized system is required, and this is another problem that motivates the research of efficient methods to recover the eigenvalues of the multibody system [6,7].

Motivated by the above-mentioned applications, in this paper, we discuss the numerical difficulties related to the computation of eigenvalues and eigenvectors in multi-flexible-body systems under the most general assumptions: we assume that the system can present singular modes (also called rigid body or free–free modes); we consider the optional presence of damping, hence leading to complex-valued eigenpairs; we consider an arbitrary number of parts and constraints; and we assume that the size of the system could be arbitrarily large. In particular, this last requirement imposes some limitations on the type of solver that must preserve the sparsity of the matrices for the sake of an acceptable computational performance and that should be able to output just a small subset of eigenvalues, either the lowest ones or those clustered around a frequency of interest.

The problem of the eigenvalue computation in multibody systems is discussed by various authors in the literature, although the topic is more common in the field of the finite

element analysis (FEA). A difficulty of multibody systems with respect to a conventional FEA is that constraints are ubiquitous and often described by algebraic equations and Lagrange multipliers. A classical approach is to remove constraints by means of an orthogonal complement that reduces the generalized coordinates to the lowest amount possible, as discussed, for instance, in [8,9]. This idea has the benefit that the linearized equations are those of an unconstrained system; thus, a conventional eigenvalue solver can be applied. However, there are also drawbacks that will be discussed in the next paragraphs.

Alternatively, one can solve an eigenvalue problem paired with constraints, thus leading to matrices that are larger but sparser. This approach is shown, for example, in [10,11]. Despite the increment in the number of unknown eigenvalues and the increment in the dimension of eigenvectors, we experienced that this approach leads to a simpler formulation. Most important, we noticed that this method preserves the sparsity of the matrices so that we could design a solver that can leverage this useful property.

An eigenvalue solver that achieved big popularity in the past years is the Implicitly Restarted Arnoldi Method (IRAM) [12]. In fact, this is the method implemented in ARPACK, a widespread Fortran77 library that can solve generalized eigenvalue problems, with both sparse or structured matrices [13]. As such, the IRAM would be sufficient to satisfy our requirements; however, we experienced that it fails to provide good convergence in some difficult cases, so we pointed our attention to the more recent Krylov–Schur method.

The Krylov–Schur method was presented in [14] as an improvement over previous Krylov subspace methods, such as the IRAM and Lanczos. Because of an efficient and robust restarting strategy, it is often able to converge even in cases where the IRAM stalls, and in general, it exhibits a superior robustness and faster convergence [15]. For these reasons, the Krylov–Schur method has become the default for the MATLAB `eigs` command, and it is also available in the SLEPC library [16], an extension of the PETSC linear algebra library, as well as in the TRILINOS library [17]. However, both are large libraries that target supercomputing and require complex build toolchains. On the other hand, there are efforts such as the SPECTRA C++ library [18] that are lightweight but might not offer all the desired functionalities, for instance, SPECTRA contains the Krylov–Schur method in a partially implemented form, making it usable only for symmetric matrices (hence the complex eigenvalue problem is out of reach, making it unuseful for a damped eigenmode computation at the moment of writing). The lack of reliable, complete and lightweight open-source libraries for computing eigenvalues with the Krylov–Schur method motivated us to develop our C++ version of it, which is described in the following pages.

In the next section, we will discuss how to obtain the needed matrices from a linearization of the multibody system; then we will review different formulations for expressing the eigenvalue problem, with or without constraints, with or without damping; then we will discuss some computational aspects related to the implementation of the sparsity-preserving Krylov–Schur solver; and finally, we will show some applications and benchmarks.

## 2. Linearization of Multibody Structures

We introduce the semi-explicit Differential Algebraic Equations (DAE) of a generic, nonlinear multi-flexible-body system with generalized coordinates $q \in \mathbb{R}^n$:

$$\begin{cases} M(q)\ddot{q} + C_q(q,t)^T \gamma = f(q,\dot{q},t) - f_g(\dot{q}) & (1) \\ C(q,t) = 0 & (2) \end{cases}$$

where $C(q,t) = 0$ is a vector of $m$ holonomic-rheonomic constraints with an $m \times n$ sparse Jacobian $C_q(q,t) = \frac{\partial C(q,t)}{\partial q}$. Moreover, $f$ is the vector of external and internal forces, and $f_g$ represent the gyroscopic and centrifugal components of the inertial forces (the full inertial forces are in fact $f_i = M\ddot{q} + f_g$).

By rewriting Equation (1) into a multivariate function

$$F(\ddot{q}, \dot{q}, q, t) = M(q)\ddot{q} + C_q(q,t)^T \gamma - f(q,\dot{q},t) + f_g(\dot{q}) = 0 \qquad (3)$$

it is easier to see that any *feasible* infinitesimal variation of the unknowns $\delta\ddot{q}$, $\delta\dot{q}$, $\delta q$ will still be of equilibrium, thus leading again to a zero-valued function, i.e., $F(\ddot{q} + \delta\ddot{q}, \dot{q} + \delta\dot{q}, q + \delta q, t) = 0$. No variation takes place on the time variable. This leads to the trivial conclusion that any step along the total derivative of the *F* function does not lead to any variation of *F*, i.e.,

$$\begin{bmatrix} \delta\ddot{q} & \delta\dot{q} & \delta q \end{bmatrix} \begin{bmatrix} \frac{\partial F}{\partial\ddot{q}} & \frac{\partial F}{\partial\dot{q}} & \frac{\partial F}{\partial q} \end{bmatrix}^T \bigg|_{\ddot{q},\dot{q},q,t} = \mathbf{0} \tag{4}$$

By expanding the evaluation of the partial derivatives of *F* to all the terms contained in it the following result is obtained where all the derivatives of the forces have been meaningfully collected into more readable stiffness *K* and damping *R* matrices:

$$\begin{cases} M(q)\delta\ddot{q} + R(q,\dot{q})\delta\dot{q} + K(q,\dot{q},\ddot{q},\gamma)\delta q + C_q(q,t)^T\delta\gamma = \mathbf{0} & (5) \\ C_q(q,t)\delta q = \mathbf{0} & (6) \end{cases}$$

In the formula above, the damping matrix *R* comes from the linearization of internal/external forces $f$ about $\dot{q}$, plus the linearization of $f_g$, the quadratic part of the inertial forces; hence,

$$R(q,\dot{q}) = -\frac{\partial f(q,\dot{q},t)}{\partial\dot{q}} + \frac{\partial f_g(\dot{q})}{\partial\dot{q}} \tag{7}$$

$$= R_f + R_i \tag{8}$$

Note that the $R_i$ part also includes the so-called *gyroscopic damping*, and it is null for $\dot{q} = \mathbf{0}$.

The tangent stiffness *K* contains the effect of the linearization of internal and external forces ($K_f$ i.e. the conventional stiffness matrix), the linearization of the inertial forces ($K_i$)—that is null if, as often happens, the system is studied in a static configuration, but might be relevant otherwise when studying, for example, eigenmodes of a rotating wind turbine—and the linearization of the constraint reaction forces $K_c$. It can be noted that the latter can introduce a contribution to the tangent stiffness due to the geometric effect of changes in $C_q(q,t)$ about the linearization point. One example is offered by the gravity-induced stiffness of a pendulum, where the rotation of the pendulum generate changes in $C_q$ due to the change of the reaction force at the pendulum hinge. If the other sources of stiffness are more relevant (e.g., springs, elastic internal forces in beams, etc.) or if $\lambda$ is small at the linearization point, then this term might be neglected.

Because of these reasons, a static or dynamic analysis should be performed right before computing eigenvectors, since the value of $\gamma$ must be known when computing (9):

$$K(q,\dot{q},\ddot{q},\gamma) = \frac{\partial(M(q)\ddot{q} + f_g)}{\partial q} + \frac{\partial(C_q(q,t)^T\gamma)}{\partial q} - \frac{\partial f(q,\dot{q},t)}{\partial q} \tag{9}$$

$$= K_i + K_c + K_f \tag{10}$$

Oftentimes, especially in the FEA literature, the $K_f$ matrix is split in two components $K_f = K_{f_m} + K_{f_g}$, where $K_{f_m}$ is the material stiffness and $K_{f_g}$ is the geometric stiffness—the latter is caused, for example, by change in orientation of internal forces in beams, and its effect is null in configurations that have no initial stress at the linearization point.

A further splitting can be performed by distinguishing internal forces, caused by finite elements, and external forces, caused by applied loads; thus, $f(q,\dot{q},t) = f(q,\dot{q},t)_{int} + f(q,\dot{q},t)_{ext}$, and $K_f = K_{f_m int} + K_{f_g int} + K_{f_m ext} + K_{f_g ext}$. In many cases, $K_{f ext}$ matrices are of small value if compared to $K_{f int}$ matrices and can be neglected, but in other cases, for example, when considering aerodynamic loads, they might be relevant.

We remark that (5) and (6) require the introduction of constraints via Jacobian matrices $C_q(q,t)$ and Lagrange multipliers $\delta\gamma$: in fact, in the following, we will handle this complication by solving *constrained* eigenvalue problems. However, one might wonder if there

is an alternative approach that avoids $C_q(q, t)$ and $\delta\gamma$ at all, so that a conventional (not constrained) eigenvalue solver could be used. Actually, this would be possible, for example, by running a QR decomposition on the $C_q$ matrix in order to find a $\Xi \in \mathbb{R}^{n \times m}$ matrix such that $\Xi^T C_q(q, t)^T = 0$. In this way, one could introduce a smaller set of independent variables $y \in \mathbb{R}^{n-m}$ for whom $\dot{q} = \Xi\dot{y}$, hence rewriting the DAE (1) as a simple ODE

$$\Xi^T M(q)\Xi\ddot{y} + \Xi^T M(q)\dot{\Xi}\dot{y} = \Xi^T f(q, \dot{q}, t) - \Xi^T f_g(\dot{q}) \tag{11}$$

This can be linearized to give a single expression which is an alternative to (5) and (6):

$$M_Y(q)\delta\ddot{y} + R_Y(q, \dot{q})\delta\dot{y} + K_Y(q, \dot{q}, \ddot{q})\delta y = 0 \tag{12}$$

with

$$M_Y(q) = \Xi^T M(q)\Xi \tag{13}$$

$$R_Y(q) = -\Xi^T \frac{\partial f(q, \dot{q}, t)}{\partial \dot{y}} + \Xi^T \frac{\partial f_g(\dot{q})}{\partial \dot{y}} + \Xi^T M(q) \frac{\partial \dot{\Xi}\dot{y}}{\partial \dot{y}} \tag{14}$$

$$K_Y(q, \dot{q}, \ddot{q}) = \left( \frac{\partial \Xi^T}{\partial y} M(q)\Xi + \Xi^T \frac{\partial M}{\partial y}(q)\Xi + \Xi^T M(q)\frac{\partial \Xi}{\partial y} \right)\ddot{y} \tag{15}$$

$$+ \left( \frac{\partial \Xi^T}{\partial y} M(q)\dot{\Xi} + \Xi^T \frac{M(q)}{\partial y}\dot{\Xi} \right)\dot{y}$$

$$- \frac{\partial \Xi^T}{\partial y} f(q, \dot{q}, t) - \Xi^T \frac{\partial f(q, \dot{q}, t)}{\partial y} + \frac{\partial \Xi^T}{\partial y} f_g(\dot{q}) + \Xi^T \frac{\partial f_g(\dot{q})}{\partial y}$$

However, we note that the expressions of $R_Y$ and $K_Y$ are substantially more intricate than the expression of $R$ and $K$ in (7) and (9), especially considering that (15) would require the knowledge of $\dot{\Xi}$ and $\partial\Xi^T/\partial y$.

While these latter terms might be neglected in some cases—thus reducing matrices to the approximated forms $R_Y(q) \approx \Xi^T R\Xi$, $K_Y(q) \approx \Xi^T K\Xi$—we experienced that such a simplification is possible only when constraints do not change direction in a significant way: in fact, even a simple example of an oscillating pendulum would erroneously give zero natural frequency with this simplification.

Moreover, the multiplications by $\Xi$ and $\Xi^T$ will destroy the sparsity of the original matrices $M$, $R$, $K$: this is not an issue in problems of small size, but for large problems this would lead to unacceptable memory and performance requirements.

For these reasons, we prefer to proceed with the linearization expressed in (5) and (6) at the cost of dealing with constraints during the iterative eigenvalue solution process. The following section will explain how to use the $M$, $R$, $K$, $C_q$ matrices to this end.

## 3. Modal Analysis

We can distinguish two types of modal analysis: in the first case, we search for real-valued eigenvalues of the undamped system; in the second case, we search for complex-valued eigenvalues of the damped system. The former can be considered a subcase of the latter for $R = 0$, and hence a single solver could attack both problems; however, it is better to adopt two different solution schemes in order to exploit some optimizations that lead to a high computational performance if the damping is of no interest.

### 3.1. Undamped Case—Real Valued

We recall some basic concepts in eigenvalue analysis of dynamic systems.

For the simple case of an unconstrained, undamped system with $R = 0$, with solutions $q = \Sigma(\Phi_i e^{i\omega t} + \Phi_i e^{-i\omega t})$

$$M\ddot{q} + Kq = 0 \tag{16}$$

it is possible to compute the eigenmodes from the following characteristic expression:

$$\left(-\omega_i^2 M + K\right)\mathbf{\Phi}_i = \mathbf{0} \tag{17}$$

that leads to a standard eigenvalue problem (SEP) with eigenvalues $\lambda_i = \omega_i^2$ and matrix $C = M^{-1}K$:

$$\left(M^{-1}K - \lambda_i I\right)\mathbf{\Phi}_i = \mathbf{0} \tag{18}$$

$$(C - \lambda_i I)\mathbf{\Phi}_i = \mathbf{0} \tag{19}$$

For symmetric $K$ and $M$, by the spectral theorem, eigenpairs are real.

However, there are some difficulties that prevent the direct use of (18) in engineering problems of practical interest:

- It works only if there are no constraints (no $C_q$ Jacobian matrix);
- It requires the inversion of the $M$ matrix: even if $M$ is often diagonal-dominant and easy to invert, this is not true in general, and it could destroy the sparsity of the matrices in the case of large systems;
- We may be interested in just a small subset of eigenvalues, usually the lower modes, so we need an iterative scheme that is able to do this.

We compute the modes of the *constrained undamped* multibody system with the following *generalized eigenvalue problem* (GEP):

$$-\begin{bmatrix} K & C_q^T \\ C_q & 0 \end{bmatrix}\hat{\mathbf{\Phi}}_i = \lambda_i \begin{bmatrix} M & 0 \\ 0 & 0 \end{bmatrix}\hat{\mathbf{\Phi}}_i \tag{20}$$

where we introduce the augmented eigenvector

$$\hat{\mathbf{\Phi}}_i = \{\mathbf{\Phi}_i^T, \boldsymbol{\xi}_i^T\}^T.$$

and where we recover natural frequencies as:

$$\omega_i = \sqrt{-\lambda_i}, \quad f_i = \omega_i/2\pi \tag{21}$$

We remark that one could change the sign in the left-hand side of (20); this would obtain positive eigenvalues and then one would compute $\omega_i = \sqrt{\lambda_i}$ instead.

The solution of the problem (20) generates $n + m$ eigenvalues, where only $n$ is of interest, and $m$ is spurious modes with $\lambda \approx \pm\infty$ that can be discarded. The same filtering must be performed for the corresponding eigenvectors. Moreover, the last $m$ components of the eigenvectors, namely $\boldsymbol{\xi}_i$, can just be discarded or used to get insight about reaction forces because they represent the amplitude of reactions in constraints during the periodical motion of the system.

Alternatively, one can solve

$$-\begin{bmatrix} K & 0 \\ 0 & 0 \end{bmatrix}\hat{\mathbf{\Phi}}_i = \lambda_i \begin{bmatrix} M & C_q^T \\ C_q & 0 \end{bmatrix}\hat{\mathbf{\Phi}}_i \tag{22}$$

but this would produce $m$ spurious modes with $\lambda \approx 0$ that can easily be confused with those eigenvalues resulting from rigid body modes. These latter, also known as *free-free modes*, result from bodies that retain some unconstrained degree of freedom, that turn out to have $\lambda_i \approx 0$ too.

In this formulation (22), the last $m$ components of the eigenvectors, namely $\boldsymbol{\xi}_i$, represent the second integration of reaction forces/moments of the constraints, which can be discarded because no physical meaning exists.

The matrices that appear in the two forms of the GEP have different properties, and this is relevant when we will choose the optimal solution scheme. In the GEP (20), the $A$ matrix

is nonsingular only if there are no rigid body modes, as it is *z*-times rank deficient in the presence of *z* rigid body modes. Moreover, the *B* matrix is always singular and not invertible. Hence, both matrices are not invertible in the most general case. On the other hand, in the GEP (22), the *A* matrix is singular, but the *B* matrix is always nonsingular and invertible, regardless of the presence of rigid body modes, because *M* is positive definite and $C_q$ is assumed to be full rank. This would make GEP (22) a better choice with respect to GEP (20) because one could always transform it to an SEP via $C = B^{-1}A$. However, as we will see later, solving the SEP in this form is not what we need in the case of large systems, where we want a limited number of eigenpairs starting from the smallest ones. If so, a shift-and-invert approach is needed, where the nonsingularity of *B* is irrelevant, and we would rather need the inversion of *A*. In this case, neither GEP (20) nor GEP (22) would fit this requirement. However, the shift-and-invert approach requires a regularized form of the inverse matrix, by means of a $\sigma$ shift parameter as in $C = (A - \sigma B)^{-1}B$, so both approaches could work in this setting, except for $\sigma = 0$.

Finally, we note that, when the *K* matrix is symmetric, both *A* and *B* are symmetric; therefore, optimized linear solvers for the inner loop of the Krylov–Schur solver could be used for the sake of a higher speed (that is, the $(A - \sigma B)^{-1}$ problem can be approached via LDLt decompositions rather than LU decompositions in the case of direct solvers, or via the MINRES rather than the GMRES in the case of Krylov solvers).

### 3.2. Damped Case—Complex Valued

The conventional modal analysis of the damped system

$$M\ddot{q} + R\dot{q} + Kq = 0 \tag{23}$$

with solutions $q = \Phi e^{\lambda t}$ formulated as a quadratic eigenvalue problem (QEP), either with left or right eigenvectors:

$$(\lambda^2 M + \lambda R + K)\Phi = 0 \tag{24}$$

$$\Psi^*(\lambda^2 M + \lambda R + K) = 0 \tag{25}$$

We recall some useful properties. Because coefficients of (24) are real, any complex roots must appear as complex conjugate pairs. The QEP generates 2*n* eigenvalues that are finite if *M* is nonsingular; if *M*, *R*, *K* are real, or Hermitian, then eigenvalues can be a mix of real values or complex conjugate pairs $(\lambda, \overline{\lambda})$; if *M* is Hermitian positive definite and *R*, *K* are Hermitian positive semidefinite, then $\text{Re}(\lambda) \leq 0$.

- Complex conjugate pairs $(\lambda, \overline{\lambda})$ correspond to underdamped modes, oscillatory and decaying for $\text{Re}(\lambda) < 0$;
- Purely imaginary conjugate pairs $(\lambda, \overline{\lambda})$, $\text{Re}(\lambda) = 0$ correspond to undamped modes, purely harmonic and not decaying;
- Real modes with $\text{Re}(\lambda) \leq 0$ and no imaginary part correspond to overdamped modes, not oscillatory, exponential decaying;
- In all cases, $\text{Re}(\lambda) > 0$ indicates an unstable system;
- For the class of damped systems, also eigenvectors $\Phi_i$ are complex, with elements:

$$\Phi_{i,j} = a_{i,j} + i\, b_{i,j} = \delta_{i,j} e^{i\beta_{i,j}}$$

  where both the amplitude and the phase of the entire eigenvector can be arbitrary (but the relative amplitude $\delta_{i,j}/\delta_{i,k}$ of each component is unaltered by whatever normalization, and the relative phase of each component is constant $\beta_{i,j} - \beta_{i,k} = \text{const}_{jk}$);
- The two eigenvectors of a complex conjugate pair are also conjugate.

Oscillatory modes, corresponding to a complex conjugate pair $(\lambda, \overline{\lambda})$, $\text{Re}(\lambda) < 0$, can be written in a more engineering-oriented way as done in 1-dof systems, $Ae^{(-\zeta\omega + i\omega_d)t} + Be^{(-\zeta\omega - i\omega_d)t}$,

where one has the following expressions for natural (undamped) frequencies $\omega_i$, damped frequencies $\omega_{d,i}$ and damping factors $\zeta_i$:

$$\omega_i = \|\lambda_i\|, \quad f_i = \omega_i/2\pi \tag{26}$$

$$\omega_{d,i} = \text{Im}(\lambda_i), \quad f_{d,i} = \omega_{d,i}/2\pi \tag{27}$$

$$\zeta_i = -\text{Re}(\lambda_i)/\omega_i \tag{28}$$

$$\omega_{d,i} = \omega_i\sqrt{1-\zeta^2} \tag{29}$$

Although there exist algorithms that can solve (24) directly, often the QEP is transformed to an SEP or GEP so that a conventional solver like Arnoldi or Krylov–Schur can be used. This can be performed by expressing the problem in state space: we introduce an augmented eigenvector that contains both the eigenvector $\boldsymbol{\Phi}_i \in \mathbb{R}^n$ and the eigenvector $\lambda_i \boldsymbol{\Phi}_i \in \mathbb{R}^n$:

$$\underline{\boldsymbol{\Phi}}_i^T = \{\boldsymbol{\Phi}_i^T, \lambda_i \boldsymbol{\Phi}_i^T\}$$

This can be used to transform the QEP (24) into the following GEP with double the original size:

$$\begin{bmatrix} 0 & I \\ -K & -R \end{bmatrix} \underline{\boldsymbol{\Phi}}_i = \lambda_i \begin{bmatrix} I & 0 \\ 0 & M \end{bmatrix} \underline{\boldsymbol{\Phi}}_i \tag{30}$$

Additionally, one can consider the constraints by introducing Lagrange multipliers $\boldsymbol{\xi}_i \in \mathbb{R}^m$ that correspond to the $m$ constraints enforced as $C_q \boldsymbol{\Phi}_i = 0$, thus obtaining a constrained QEP:

$$\begin{cases} \lambda_i^2 M \boldsymbol{\Phi}_i + \lambda_i R \boldsymbol{\Phi}_i + K \boldsymbol{\Phi}_i + C_q^T \boldsymbol{\xi}_i = \mathbf{0} & (31) \\ - C_q \boldsymbol{\Phi}_i = \mathbf{0} & (32) \end{cases}$$

Finally, introducing the augmented eigenvector $\underline{\hat{\boldsymbol{\Phi}}}_i \in \mathbb{R}^{2n+m}$ as

$$\underline{\boldsymbol{\Phi}}_i^T = \{\boldsymbol{\Phi}_i^T, \lambda_i \boldsymbol{\Phi}_i^T, \boldsymbol{\xi}_i^T\}$$

and by making use of simple linear algebra, we can write the constrained QEP as a constrained GEP:

$$\begin{bmatrix} 0 & I & 0 \\ -K & -R & -C_q^T \\ -C_q & 0 & 0 \end{bmatrix} \underline{\hat{\boldsymbol{\Phi}}}_i = \lambda_i \begin{bmatrix} I & 0 & 0 \\ 0 & M & 0 \\ 0 & 0 & 0 \end{bmatrix} \underline{\hat{\boldsymbol{\Phi}}}_i \tag{33}$$

An alternative formulation is based on the solution of the following GEP, where the spurious modes related to the constraint equations are zero instead of infinite:

$$\begin{bmatrix} 0 & I & 0 \\ -K & -R & 0 \\ 0 & 0 & 0 \end{bmatrix} \underline{\hat{\boldsymbol{\Phi}}}_i = \lambda_i \begin{bmatrix} I & 0 & 0 \\ 0 & M & C_q^T \\ C_q & 0 & 0 \end{bmatrix} \underline{\hat{\boldsymbol{\Phi}}}_i \tag{34}$$

that corresponds to

$$\begin{cases} \lambda_i^2 M \boldsymbol{\Phi}_i + \lambda_i R \boldsymbol{\Phi}_i + K \boldsymbol{\Phi}_i + C_q^T \lambda_i \boldsymbol{\xi}_i = \mathbf{0} & (35) \\ C_q \lambda_i \boldsymbol{\Phi}_i = \mathbf{0} & (36) \end{cases}$$

We experienced that, among the different formulations (Table 1), the most efficient way to compute eigenpairs for the constrained damped system is the GEP approach (33).

**Table 1.** Different options for the eigenpair computation.

| | GEP | | Notes |
|---|---|---|---|
| Undamped | $A = [K] \quad B = [M]$ | | real eigenpairs, $\omega_i = \sqrt{\lambda_i}$<br>$A$ singular if rigid body modes |
| | $A = [-K] \quad B = [M]$ | | real eigenpairs, $\omega_i = \sqrt{-\lambda_i}$<br>$A$ singular if rigid body modes |
| Undamped Constrained | $A = \begin{bmatrix} -K & -C_q^T \\ -C_q & 0 \end{bmatrix} \quad B = \begin{bmatrix} M & 0 \\ 0 & 0 \end{bmatrix}$ | | real eigenpairs, $\omega_i = \sqrt{-\lambda_i}$<br>$\|\lambda_i\| = \infty$ for each constraint<br>$A$ singular if rigid body modes<br>$B$ singular |
| | $A = \begin{bmatrix} -K & 0 \\ 0 & 0 \end{bmatrix} \quad B = \begin{bmatrix} M & C_q^T \\ C_q & 0 \end{bmatrix}$ | | real eigenpairs, $\omega_i = \sqrt{-\lambda_i}$<br>$\lambda_i = 0$ for each constraint<br>$A$ singular<br>$B$ nonsingular |
| Damped | $A = \begin{bmatrix} 0 & I \\ -K & -R \end{bmatrix} \quad B = \begin{bmatrix} I & 0 \\ 0 & M \end{bmatrix}$ | | complex eigenpairs, $\omega_i = \|\lambda_i\|$<br>$A$ singular if rigid body modes<br>$B$ singular |
| Damped Constrained | $A = \begin{bmatrix} 0 & I & 0 \\ -K & -R & -C_q^T \\ -C_q & 0 & 0 \end{bmatrix} \quad B = \begin{bmatrix} I & 0 & 0 \\ 0 & M & 0 \\ 0 & 0 & 0 \end{bmatrix}$ | | complex eigenpairs, $\omega_i = \|\lambda_i\|$<br>$\|\lambda_i\| = \infty$ for each constraint<br>$A$ singular if rigid body modes<br>$B$ singular |
| | $A = \begin{bmatrix} 0 & I & 0 \\ -K & -R & 0 \\ 0 & 0 & 0 \end{bmatrix} \quad B = \begin{bmatrix} I & 0 & 0 \\ 0 & M & C_q^T \\ C_q & 0 & 0 \end{bmatrix}$ | | complex eigenpairs, $\omega_i = \|\lambda_i\|$<br>$\lambda_i = 0$ for each constraint<br>$A$ singular<br>$B$ nonsingular |

## 4. Computing Eigenpairs with Sparse Matrices

When the number of unknowns $n$ grows, it is not possible to compute all the $n$ eigenvalues and eigenvectors, both for reasons of computational time and for the extreme requirement of the memory needed for storing all the eigenvectors. In fact, many analyses that require the computation of eigenmodes in practice require a small set of them.

There are iterative methods that preserve the sparsity of matrices and that can compute a limited set of $k$ eigenvectors: most notably, these are the IRAM (Implicitly Restarted Arnoldi Method), the Locally Optimal Block Preconditioned Conjugate Gradient (LOBPCG) and, lastly, the Krylov–Schur method.

The problem is that they compute the largest $k$, not the smallest ones, which is exactly the opposite of our interest. This issue can be solved adopting a Moebius transform of the eigenvalue problem. We proceed as follows.

For the undamped constrained case, first we formulate the generalized eigenvalue problem (GEP):

$$A\hat{\boldsymbol{\Phi}}_i = \lambda_i B \hat{\boldsymbol{\Phi}}_i \tag{37}$$

$$A = \begin{bmatrix} -K & -C_q^T \\ -C_q & 0 \end{bmatrix} \tag{38}$$

$$B = \begin{bmatrix} M & 0 \\ 0 & 0 \end{bmatrix} \tag{39}$$

then we adopt a Moebius transform of the eigenvalue problem, namely the *shift-and-invert* strategy that computes eigenvalues $\mu_i$ in the following problem:

$$(C - \mu_i I)\hat{\boldsymbol{\Phi}}_i = 0 \tag{40}$$

$$C = (A - \sigma B)^{-1} B \tag{41}$$

$$\mu = \frac{1}{\lambda - \sigma} \quad \lambda = \frac{1}{\mu} + \sigma \tag{42}$$

After the eigenvalue problem (40) is solved for $k$ pairs of $(\mu_i, \hat{\boldsymbol{\Phi}}_i)$, one recovers the original $\lambda_i$ and hence the original $\omega_i$ using (42).

For the damped constrained case, we formulate a GEP of the type

$$A\hat{\underline{\boldsymbol{\Phi}}}_i = \lambda_i B\hat{\underline{\boldsymbol{\Phi}}}_i \tag{43}$$

$$A = \begin{bmatrix} 0 & I & 0 \\ -K & -R & -C_q^T \\ -C_q & 0 & 0 \end{bmatrix} \tag{44}$$

$$B = \begin{bmatrix} I & 0 & 0 \\ 0 & M & 0 \\ 0 & 0 & 0 \end{bmatrix} \tag{45}$$

then, similarly to the undamped case, we apply the shift–invert Moebius transformation to solve $(C - \mu_i I)\hat{\underline{\boldsymbol{\Phi}}}_i = 0$ with $C = (A - \sigma B)^{-1}B$, obtaining pairs $(\mu_i, \hat{\underline{\boldsymbol{\Phi}}}_i)$, and finally recovering $\lambda = \frac{1}{\mu} + \sigma$.

Right eigenvectors $\boldsymbol{\Phi}_i$ are not affected by the Moebius transform. Just in case one is interested in the left eigenvectors as in $\boldsymbol{\Psi}_i^*(C - \lambda_i I) = 0$, then those are recovered solving $z_i^*(C - \mu_i I) = 0$ and using the transform $z = (A - \sigma B)^*\boldsymbol{\Psi}_i$.

User-defined values of $\sigma$ can be used to extract eigenvalues in specific frequency ranges. In fact, the iterative solver will return the $k$ eigenvalues that are closer, in absolute value, to $\sigma$.

If the shift parameter $\sigma$ is zero or close to it, as often happens, one can see that the largest $k$ eigenvalues $\mu$ computed by the Krylov–Schur solver will become the smallest $k$ eigenvalues $\lambda$, for the modes closer to zero frequency.

As a special case, for $\sigma = 0$, one has $C = A^{-1}B$ and $\lambda = \frac{1}{\mu}$, that for an unconstrained problem (no $C_q$ Jacobian) it corresponds to solving the inverse eigenvalue problem $(K^{-1}M - \mu_i I)\boldsymbol{\Phi}_i = \mathbf{0}$.

In general, one can adjust the $\sigma$ shift value so that it provides the best numerical performance; in detail, it provides a regularization of $A$ and helps solve the linear problem in (41) and also in the case where $A$ or $B$ are singular or close to singularity. This is what happens in many cases when conducting a modal analysis of engineering structures, especially if the structure has rigid body modes. In fact, our default method is to extract all the lower modes, including rigid body modes, and at this end, we experienced that a value of $\sigma = 1 \times 10^{-3}$ works well also to retrieve the six $\lambda \approx 0$ modes and to cure ill-posed problems.

The Krylov–Schur and Arnoldi methods draw on a single computational primitive, that is, the product of a sparse matrix $C$ by a vector $v$ for the solution of the problem $(C - \mu_i I)\boldsymbol{\Phi}_i = 0$. However, in our case, $C = (A - \sigma B)^{-1}B$; hence, pre-computing such a $C$ matrix is out of question because the exact inversion of $(A - \sigma B)$ would require too much CPU time and would destroy the sparsity. Because only the product primitive $Cv$ is required for the iterative solver, an acceptable trade-off is to return the result of the product $r = Cv$ by performing these steps:

$$z = Bv \tag{46a}$$

$$r = (A - \sigma B)^{-1}z \tag{46b}$$

Here, we note that Equation (46b) in the second step requires a linear system solution. This can be a computational bottleneck, but a substantial speedup can be achieved, observing that the coefficient matrix $(A - \sigma B)$ is constant; therefore, one can factorize it once at the beginning of the Krylov–Schur iterations and perform only the back substitutions in (46b).

An alternative that preserves the sparsity of the matrices and can fit better in scenarios with millions of unknowns is that (46b) is solved iteratively via truncated MINRES or

GMRES iterative methods. If the number of unknowns is in the range of tens of thousands, however, we experienced that the factorization via a direct method performs faster.

## 5. Implementation of the Krylov–Schur Solver

The Krylov–Schur method was introduced in 2001 [14], leading to an improved performance in respect to other Krylov subspace methods, such as Arnoldi and Lanczos, which were used for decades in the field of eigenvalue computation. The notorious Implicitly Restarted Arnoldi Method, implemented, for example, in the ARPACK library, or the Locally Optimal Block Preconditioned Conjugate Gradient (LOBPCG), implemented, for example, in the BLOPEX library, both fail to converge for those problems whose matrix is of type (20), (22), (33) or (34) and for which wide mass ratios or strongly ill-conditioned blocks are present.

On the contrary, the robustness of the Krylov–Schur method also guarantees satisfying results for the most critical conditions, thus becoming the elected choice for the following tests. Our implementation follows the guidelines in [19] as reported in Algorithm 1. It was extended to the case of complex and sparse matrices and is included in the open-source multibody library CHRONO [20].

On a parent level of the Krylov–Schur solver, specific routines construct an eigenvalue problem in accordance with (33) or with (20) for the undamped case. This will push the spurious constraint modes to infinity, being of less disturbance for the usual low-frequency area of interest for engineering applications.

The code offers the possibility to specify different problem formulations, either direct or in shift–invert, by providing different OP_CV($v$) operators in Algorithm 2. For instance, in Algorithm 3, we show the implementation for the shift–invert case, implementing (46a) and (46b).

The solutions of the linear systems required by the method can be theoretically provided by any linear solver enabled for complex values; practically, given the relatively high accuracy required by the solution and the ill-conditioning of some matrices, direct solvers are almost mandatory for this application, relegating iterative solvers only for systems with higher degrees of freedom. However, for smaller and simpler problems, the choice of the solver is not critical (allowing the use of, e.g., SparseLU and SparseQR functions from the popular C++ linear algebra library EIGEN [21]); for most of the real cases, more advanced solvers are required, such as Pardiso MKL or MUMPS [22]. Given the importance of this choice, our Krylov–Schur implementation was made solver agnostic: the user can indeed provide one of its own choice.

For the undamped case, as in (20), the value of $\sigma$ in the shift–invert procedure is assigned as a small positive real value $\sigma = \epsilon$ (by default, we used $\sigma = 10^{-3}$ in our tests) in order to return the lowest eigenmodes, including those with zero eigenvalues in case there are rigid body modes. A small negative real value would work as well, but the numerical conditioning of the problem would be worse. If one needs specific eigenmodes clustered about some specific frequency $f_c$, we set it as $\sigma = -f_c^2$. For the damped case, we use a complex shift $\sigma = \sigma_R + i\sigma_I$, with a small real value $\sigma_R = \epsilon$ and no imaginary part if we are interested in the lowest eigenvalues, for instance, $\sigma = 10^{-3} + i0$, or with a finite imaginary part if we need eigenmodes clustered about an $f_c$ frequency: $\sigma = \epsilon_R + if_c$.

The Krylov–Schur decomposition is then solved by using the EIGEN linear algebra library eigensolvers [21].

An important contribution to the stability of the method is given by a trivial and inexpensive preconditioning of the Jacobian matrix $C_q$. While the stiffness and damping matrices usually have terms in the order of at least $10^6$, the Jacobian matrix is usually in the order of $10^0$. This change affects only the Lagrange multipliers $\gamma$ and the relative eigenvector counterpart $\xi_i$ that should be re-scaled back for the same factor (if they are of any interest to the user). This simple change in the matrices allows, in some corner case, a significant reduction in the residuals, even just after the first iteration of the method.

---

**Algorithm 1** Krylov–Schur

---

1: **procedure** KRYLOV–SCHUR(OP_CV(),$k$,$m$)
2:     $Q(:,1) := v1/norm(v1)$
3:     $p := 1$
4:     $isC := 0$
5:     $[Q,H] := $ KRYLOVEXPANSION(OP_CV(),$Q$,$H$,$0$,$k$)
6:     **while** $i < i_{\max}$  &  $p < k$ **do**
7:         $i ++$
8:         $isC := 0$
9:         $[Q,H] := $ KRYLOVEXPANSION(OP_CV(),$Q$,$H$,$k + isC$,$m$)
10:        $[U,T,isC] := $ SORTSCHUR($H(p:m,p:m)$,$k - p + 1$)
11:        $H(p:m,p:m) := T$
12:        $H(1:p-1,p:m) := H(1:p-1,p:m)\,U$
13:        $Q(:,p:m) := Q(:,p:m)\,U$
14:        $H(m+1,p:m) := H(m+1,m)\,U(end,:)$
15:        $Q := [Q(:,1:k),Q(:,m+1)]$
16:        $H := [H(1:k,1:k);H(m+1,1:k)]$
17:        CHECKCONVERGENCE($H$,$k + isC$,$p$,$tol$)
18:     **end while**
19:     $[\mu,\boldsymbol{\Phi}_H] := $ EIG($H(1:k+isC,1:k+isC)$)
20:     $\boldsymbol{\Phi} = Q(:,k+isC)\boldsymbol{\Phi}_H$
21:     **return** $\mu$, $\boldsymbol{\Phi}$
22: **end procedure**

---

**Algorithm 2** Krylov Expansion

---

1: **procedure** KRYLOVEXPANSION(OP_CV(),$Q$, $H$, $k_s$, $k_e$)
2:     **for** $k = k_s + 1 : k_e$ **do**
3:         $v := $ OP_CV($Q(:,k)$)
4:         $isC := 0$
5:         $w := Q(:,1:k)'\,v$
6:         $v -= Q(:,1:k)\,w$
7:         $w2 := Q(:,1:k)'\,v$
8:         $v -= Q(:,1:k)\,w2$
9:         $w += w2$
10:        $nv := norm(v)$
11:        $Q(:,k+1) := v/nv$
12:        $H(1:k+1,k) := [w;nv]$
13:     **end for**
14: **end procedure**

---

**Algorithm 3** Op_Cv operator

---

1: **procedure** OP_CV($v$)
2:     $z = Bv$
3:     $r = (A - \sigma B)^{-1}z$
4:     **return** $r$
5: **end procedure**

---

## 6. Results

The Krylov–Schur method was tested on various scenarios, including real-case problems, in order to assess the accuracy and scalability of the method. The relevant test conditions include flexible elements, rigid bodies, generic constraints and free–free modes in various combinations.

The tests are leveraging the newly implemented quadratic Krylov–Schur eigensolver using the Pardiso MKL direct linear solver, and the results are compared to the *eigs* solver of MATLAB (that turns out to be an implementation of the Krylov–Schur solver as well) as well as against the state-of-the-art Arpack [23] routines, making sure that—even for this latter library—the same Pardiso MKL linear solvers were used. The hardware includes an Intel i7 6700HQ with 16 GB RAM.

For the purpose of this article, only Rayleigh damping is considered. Other damping formulations can be used without expecting any drastic impact over the solver performance given that the sparsity of the matrices are kept within reasonable limits. One example that might negatively affect the solver is if *modal* damping is used: in this case, *dense* damping matrices arise, thus easily leading to increased computational costs. However, because the solver is mainly targeting sparse problems, the authors did not investigate other damping modes.

### 6.1. Hybrid Flexible and Rigid Bodies with Constraints

Constraining the system results in an additional zero-valued block in the system matrices, thus potentially compromising the stability for the inner linear solver. In the following test case, a Euler beam with properties set according to Table 2 is fixed at the base while its tip is constrained to a rigid body of a heavier mass (4000 kg) (Figure 1). The method was tested with end masses up to $10 \times 10^8$ in order to prove its robustness.

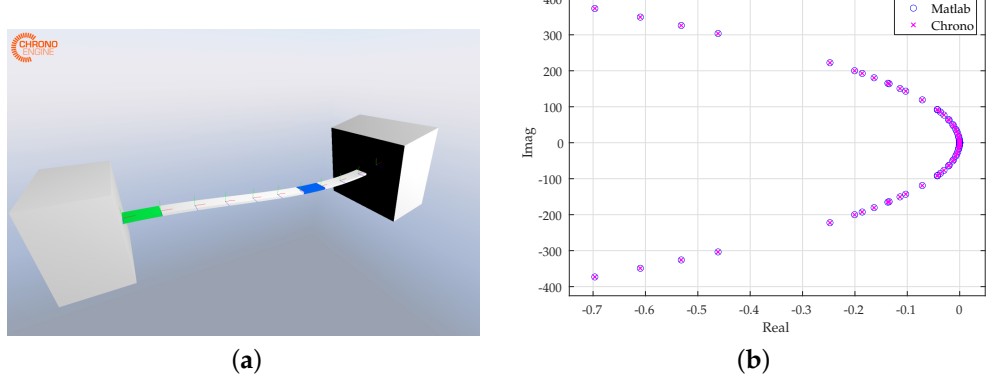

| (a) | (b) |

**Figure 1.** Cantilever Test. (**a**) Cantilever Models. (**b**) Cantilever Eigenvalues.

**Table 2.** Test beams properties.

| Property | Value |
|---|---|
| Young Modulus | 100 MPa |
| Density | $1000\,\mathrm{kg\,m^{-3}}$ |
| Section | $0.3\,\mathrm{m} \times 0.05\,\mathrm{m}$ |
| Poisson Ratio | 0.31 |
| Rayleigh Damping | $\alpha = 1 \times 10^{-3}, \beta = 1 \times 10^{-5}$ |

### 6.2. Finite Elements

An additional test case with a crank–rod–piston assembly shows the use of tetrahedral mesh (Figure 2). Without any specific preconditioner, the Arpack `dndrv4` routine failed to return consistent results in a reasonable time. The Krylov–Schur solver manifested a superior performance, especially in denser and more computationally expensive problems.

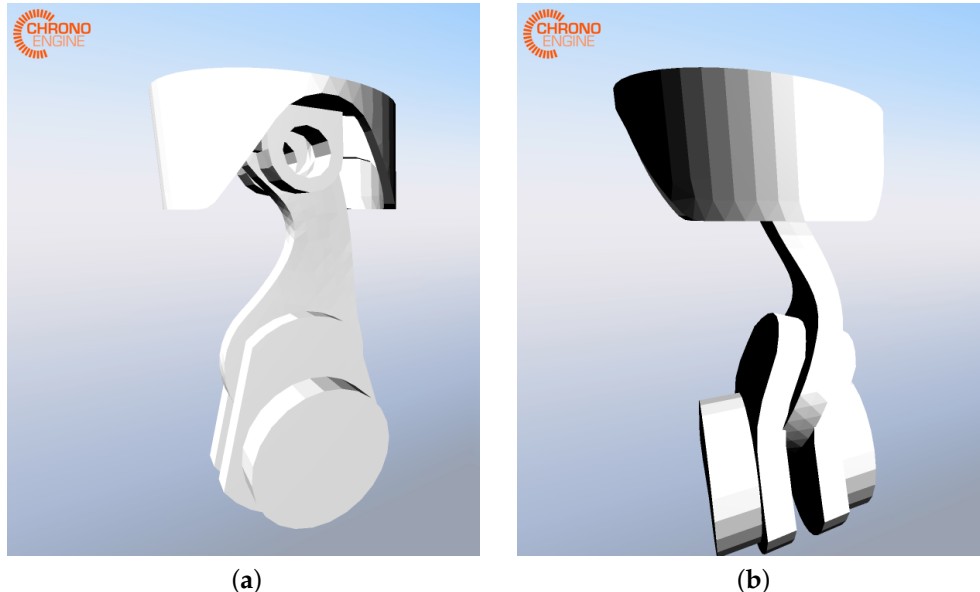

**Figure 2.** Benchmark for multibody flexible systems with constraints: flexible crank, rod and piston bodies with bearings. (**a**) Second Mode at 52 Hz. (**b**) Sixth Mode at 200 Hz.

### 6.3. Free–Free Modes

The presence of unconstrained bodies results in degenerated modes whose eigenvalues are pushed toward infinity. The method also guarantees proper stability for this degenerate case (Figure 3). It might be noticed how each degree of freedom contributes to the overall residual: the first half represents the positional degrees of freedom while the second represents the velocities (usually of less interest). The beam properties are the same as in Table 2. In this case, a comparison with Arpack revealed that, even while asking for better accuracy, the Arpack `dndrv4` routine was not able to converge to more accurate results. The Intel MKL Pardiso solver was used for both the Krylov–Schur and Arpack solvers, thus restricting the potential cause of the reduced accuracy to the eigensolver itself.

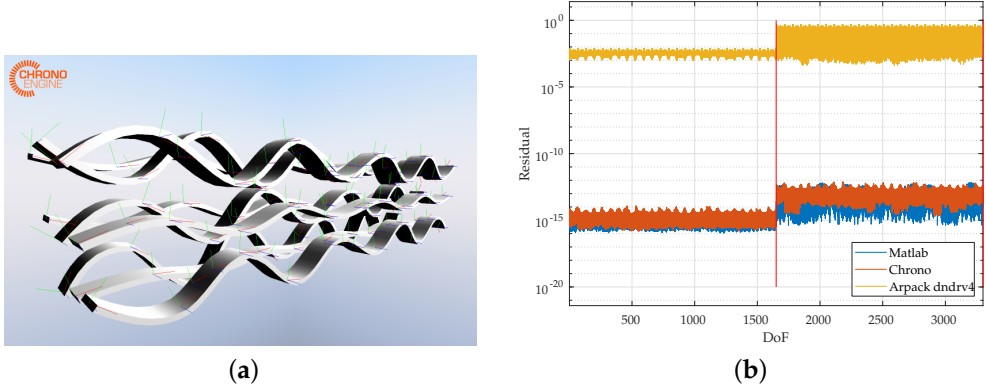

**Figure 3.** Benchmark with multiple rigid body modes. (**a**) Free–free models. (**b**) Free–free eigensolver residuals.

### 6.4. Wind Turbine

This medium-scale real test case involves a modern large-size wind turbine, courtesy of a commercial original equipment manufacturer in the wind industry. The test includes constrained flexible as well as free rigid bodies. Given the wide ratio between smaller and bigger eigenvalues (the $A$ matrix results in a reversed conditioning number of $10^{-19}$), the preconditioning of the Jacobian matrix of the constraints has proved to be essential for the robustness and accuracy of the results. The problem is non-symmetric, especially due to the linearization of the inertia ($K_i$) and constraint forces ($K_c$), as shown in Equation (9) and

more broadly in Section 2; this does not pose any additional issue to the eigensolver nor to the inner linear solver because they are both already operating on an asymmetric problem, as shown in Equation (33). Given the sensitivity of the results, the eigenvalues are not shown directly, but only the residuals, together with the stability assessment over different working conditions, are offered (Figure 4). The problem size is in the order of thousands.

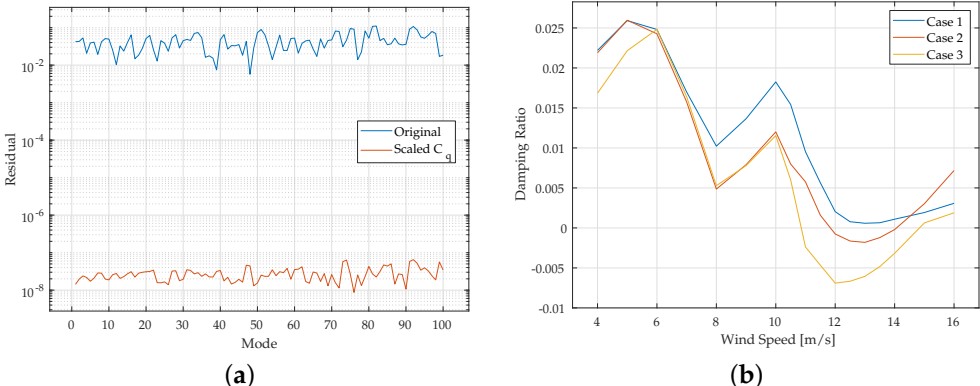

(a)                                    (b)

**Figure 4.** Wind turbine benchmark in different operating conditions. (**a**) Constraint matrix scaling effect. (**b**) Stability assessment.

### 6.5. Scalability

The scalability of the method was tested against a grid of Euler beams, whose size and number of cells are parametrized in order to provide different scales to the same problem. Each beam is fixed at every intersection with the grid. For each test, the lower 100 modes were computed. The number of elements are three and two, respectively, along each cell in the longitudinal and vertical direction. The ratio between the longitudinal and vertical number of cells is kept constant across the different tests. The results basically show a linear relation ($R^2 = 0.9996$) between the number of degrees of freedom of the original problem and the time cost of the Krylov–Schur solver, Figure 5, with a little additional overhead for smaller-scale tests. The results do not include the time expense for the assembly of the matrices. Again, the beam properties are set according to Table 2.

Moreover, in this case, the results for the Arpack solver returned high residuals, especially with close-to-zero shifts. An example over a 20 × 14 beams grid structure is shown in Figure 6.

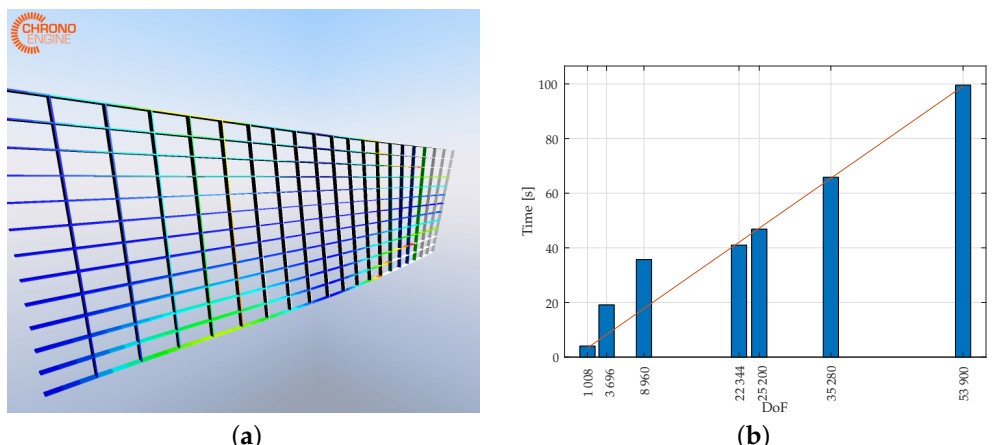

(a)                                    (b)

**Figure 5.** Scalability Test. (**a**) Beam Grid Model. (**b**) Beam Grid Time Cost.

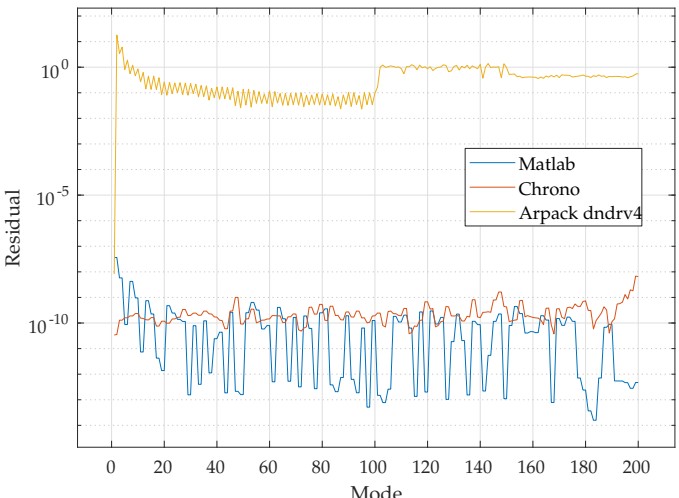

**Figure 6.** Residuals of 20 × 14 grid problem.

## 7. Conclusions

The proposed implementation of the Krylov–Schur solver successfully proves to effectively handle a wide variety of test cases, including free–free modes, constraints, rigid and flexible systems, resulting in either real or complex, symmetric or asymmetric matrices of the associated eigenvalue problem. The ample availability of the software guarantees a vast dissemination of the method, offering the best platform for further improvements.

**Author Contributions:** Conceptualization, D.M., A.T. and C.P.; methodology, D.M., A.T. and C.P.; software, D.M., A.T. and C.P.; validation, D.M., A.T. and C.P.; supervision, A.T.; funding acquisition, A.T. and C.P. All authors have read and agreed to the published version of the manuscript.

**Funding:** Part of this work has been supported by the Italian PON R&I 2014–2020 initiative (DM 1061/2021, EU cluster n.5: Climate, Energy and Mobility).

**Data Availability Statement:** The C++ version of the Krylov–Schur solver has been made available open source in the https://github.com/projectchrono/chrono repository, under BSD-3 license (accessed on 27 January 2023).

**Conflicts of Interest:** The authors declare no conflict of interest.

## Abbreviations

The following abbreviations are used in this manuscript:

| | |
|---|---|
| SEP | Standard Eigenvalue Problem |
| GEP | Generalized Eigenvalue Problem |
| QEP | Quadratic Eigenvalue Problem |

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
