# Peer review of "Complex Eigenvalue Analysis of Multibody Problems via Sparsity-Preserving Krylov–Schur Iterations"

_machines, doi:10.3390/machines11020218_

Round 1
Reviewer 1 Report
The paper deals with a deep insight on the problem of computing the eigenvalues in multibody dynamic models. At a first glance, the topic seems trivial, but it is clear that the experience of the authors clearly pointed out the pitfalls in the practical usage of standard numerical libraries.
For tese reasons, the paper is significant for the scientific multibody community.
The paper is clear and almost complete. On the other hand, to fully convince the reader of the reliability and usefulness of the implemented method, I suggest to extend and complete the numerical examples part including a comparison with other standard methods for both accuracy and computational complexity. The authors temselves state that "the method has been tested on various scenarios" but only a very few examples have been briefly discussed.
Last consideration: how does the method perform in case of not Raleight damping?
Author Response
Dear reviewer,
we went through an extensive check of our claims about the robustness of our implementation by adding plots not only of the Matlab 'eigs' solver (like in the original version of the manuscript) but also of the Arpack library (Figures 3 and 6), implementing the Implicitly Restarted Arnoldi Method and being one of the most relevant reference library.
About the Rayleigh damping we added two comments at lines 312-317, but 278-284: the advantage of Rayleigh damping over more generic formulations is to preserve the symmetry of the problem where present and to preserve the sparsity.
About symmetry: since the target of our solver is to stay as general as possible no symmetry has been leveraged (e.g. the linear solver is factorizing with LU, not LDLt). Additionally the damped problem formulation is not symmetric even if our stiffness, mass and damping matrices are.
About sparsity: unfortunately there are chances that damping matrices can become dense; that's the case, for example, for modal damping. We didn't investigate thoroughly this case especially because it lays more on the dense cases. In the article we mainly targeted sparse problems so we didn't want to move further away.
We gladly read your considerations and we hope to have answered them with appropriate care, but we are willingful to further make adjustments until a satisfying quality is achieved.
Reviewer 2 Report
The authors of this study tried to compute eigenvalues and eigenvectors of a generic multibody system dynamics by implementation of the Krylov-Schur method which results in robustness and accuracy of the problem. First a generic multibody system dynamics with holonomic-rheonomic constraints was presented, second this system was linearized, third different formulations for expressing the eigenvalue problem, with and without damping were presented, fourth the computational aspects corresponding to the sparsity-preserving Krylov-Schur solver was discussed and finally some applications were shown.
I think this work is merit and the subject of the study is of great importance, particularly in the field of control theory. Despite the inherent complexity of the problem, the method was well written and organized. However, there are some concerns that the authors need to address.
1) As mentioned in the introduction, there are other methods and algorithms can be used to compute eigenvalues like IRAM and MATLAB eign command which even uses Krylov-Schur method. The authors claim that their method has advantages in term of robustness, reliability and computational cost. To prove these claims, they need to compare the proposed method with the previous ones or provide some mathematical evidences.
2) One of the advantages for the computation of eigenvalues is to analyze the stability of dynamic systems in presence of disturbances. Is it possible to investigate or show the stability of the system that its eigenvalues were computed by the proposed method?
3) The equation of motion (Eq. 1) has a term for external forces; however, this term was not addressed when the modal analysis was explained (section 3). For both undamped and damped cases, the equations are free of external forces. Since in the abstract it has been claimed the eigenvalues were computed for the highest generality, it would be important to discuss about the effect of external forces which can be function of generalized forces.
4) To have a better understanding of the procedure related to the computation of eigenvalues, it would be proper to present and discuss the results for some simple applications like single mass-spring or single mass-spring-damper system.
Round 2
Reviewer 1 Report
The authors have responded to all the reviewer's queries satisfactorily. The paper is now suitable for publication.
Reviewer 2 Report
All my comments were addressed. I suggest to accept the manuscript.